# Too many yet too few caesarean section deliveries in Bangladesh: Evidence from Bangladesh Demographic and Health Surveys data

Md Nuruzzaman Khan[1]*, Md Awal Kabir[2], Asma Ahmad Shariff[3], Md Mostafizur Rahman[4]

1 Department of Population Science, Jatiya Kabi Kazi Nazrul Islam University, Mymensingh, Bangladesh, 2 Department of Social Work, Pabna University of Science and Technology, Pabna, Bangladesh, 3 Institute of mathematical sciences, University of Malaya, Kuala Lumpur, Malaysia, 4 Department of Population Science and Human Resource Development, University of Rajshahi, Rajshahi, Bangladesh

* mdnuruzzaman.khan@uon.edu.au

**Data Availability Statement:** BDHSs data were collected from the MEASURE DHS. The authors are restricted in sharing or making the dataset publicly available. Interested readers can download this

## Abstract

Caesarean section (CS) use is rising rapidly in Bangladesh, though lack of CS use remains common among disadvantage women. This increases risks of long-term obstetric complications as well as maternal and child deaths among disadvantage women. We aimed to determine the interaction effects of women's disadvantage characteristics on CS use in Bangladesh. For this we have analysed a total of 27,093 women's data extracted from five rounds of Bangladesh Demographic and Health Survey conducted during 2004 and 2017/18. The outcome variable was CS use, coded as use (1) and non-use (0). The major exposure variables were individual level, household level, and community level characteristics. Multilevel logistic regression model was used to determine association of CS use with socio-demographic characteristics and the interactions of three variables: working status, wealth quintile, and place of residence. We found a 751% increase of CS use over the last 13 years—from 3.88% in 2004 to 33% in 2017/18. Nearly, 80% of the total CS operation occurred in the private health facilities followed by the government health facilities (15%). Women living in rural areas with no engagement in formal income generating activities showed a 11% (OR, 0.89, 95% CI, 0.71–0.99) lower use of CS in 2004. This association was further strengthened over time, and a 51% (OR, 0.49, 0.03–0.65) lower in CS use was reported in 2017/18. Similarly, around 12%-83% lower likelihoods of CS use were found among rural poor and poorer women. These indicate Bangladesh is facing a double burden of CS use, that is a group of women with improved socio-economic condition are using this life saving procedure without medical necessity while their counterpart of disadvantage characteristics could not access the service. Improved monitoring from the government along with support to use CS services for the disadvantage groups on necessity are important.

dataset after registering with the MEASURE DHS. Necessary information are available at: http://dhsprogram.com/data/Using-DataSets-for-Analysis.cfm.

**Funding:** This research did not receive any specific grant from funding agencies in the public, commercial, or non-profit sectors.

**Competing interests:** The authors declare that they have no known competing financial interests or personal relationships that could have appeared to influence the work reported in this paper.

# Introduction

The number of caesarean section (CS) use continues to rise globally, currently accounting for 21% of all childbirths with a significant variation across countries [1]. The proportion is less than 5% in 28 countries worldwide, over three quarter of which are located in sub-Saharan Africa, including Niger, Chad, Ethiopia, and Timor Laste [2]. Only 15 countries in the world have the CS rate 10–15% [3,4], which is the World Health Organization recommendation to the significant reductions of maternal and child mortality [5]. Over 100 countries worldwide have above 15% CS use; 43 countries even have their CS use level higher than 30%. This later group is geographically heterogeneous and mostly developed countries [2]. However, recent rise on CS use rate is mainly occurring in low- and lower-middle income countries (LMICs), where improving maternal health is an ongoing challenge.

Around 42% of the total CS performed worldwide are without medical necessity, therefore, they do not have any contribution to the improvement of maternal and child health [2]. Moreover, such abusive use of CS can lead to several undesired consequences including hemorrhage and bleeding as well as associated maternal mortality and economic burden [6–8]. This is also found to be linked with a long-term loss of women's productivity as well as increasing hospitalization which further create a burden in formal healthcare delivery system [7–9].

The CS use in LMICs is generally linked to the level of development including women's education, fertility level, wealth, and contribution of the private healthcare facilities in providing CS [10,11]. Consequently, a group of women could not access this service because of financial hardship whereas another group use this service without medical necessity—an indication of the double burden of CS [11]. Moreover, along with financial capacity, geographical variation of availability of this service as well as geographical hardship to access services, such as poor transportation, could also play a significant role in differencing the CS use [12,13]. This indicates triple burden of CS—a common scenario for LMICs including Bangladesh. The underlying reasons are unavailability of formal health insurance coverage, higher rate of poverty, and rurality—the issues which are quite significant in Bangladesh [2]. This results in a major lack of access to CS use for many women in LMICs, and Bangladesh in general [2,14–17], and a few geographical regions (such as Mymensingh, Sylhet, Chattogram) in particular [10,11,13,18,19]. This contributes to the increasing maternal and child mortality which are challenges in achieving Sustainable Development Goal's target 3.1 (reduction of maternal mortality), target 3.2 (reduction of under-five mortality) and target 3.7 (ensure universal coverage of sexual and reproductive health) by 2030 [20]. However, the problem has not been addressed by previous research in many LMICs including Bangladesh where rapid rise in CS use has always been the focus of the health researchers and policy makers [14].

Researchers in LMICs including Bangladesh have explored several factors associated with the CS use and the rising trend of CS use [6,10,11,19,21]. A significant variations of CS use across factors, such as wealth quintile, education, working status, and urban/rural residence have also been identified [10,11]. However, their interaction effects on CS use have not yet been explored. Consequently, disadvantaged groups of women who could not access this life saving services are mostly unknown. We aimed to determine the interaction effects of women's disadvantage characteristics including place of residence, education, wealth quintile, working status on CS use in Bangladesh to fill the existing gap in research.

# Methods

## Study setting and design

We analysed five rounds of Demographic and Health Survey (BDHS) Data, collected in 2004, 2007, 2011, 2014 and 2017/18. The National Institute of Population Research and Training

conducted these surveys as part of the Demography and Health Survey Program of the USA. The questionnaire used for data collection was similar in all the surveys and the sampling strategy was unique, therefore they are comparable. The broad description of the survey sampling procedure is available elsewhere [22–26]. Briefly, these surveys were conducted using a multi-stage random sampling procedure.

At the first stage of the sampling, a fixed number of Primary Sampling Units (PSU) were selected randomly. The sampling frame prepared by the Bangladesh Bureau of Statistics as part of the National Population Census 2000 (for 2004, 2007 & 2011 surveys) and 2010 (for 2014 & 2017 surveys) were used to select the PSUs. Household listing operation in the selected PSUs was then carried out at the second stage of sampling. Finally, a fixed number of 30 households were selected to be included in the survey. The inclusion criteria were unique for all surveys: (i) ever married women aged 15–49 years who are usual residents of the selected households or (i) ever married women aged 15–49 years who lived the most recent night before the survey date in the selected households but not the usual residence. Selected women's reproductive data such as the number of births and the use of maternal healthcare services in the most recent pregnancy were collected. Their husbands' and under-fived aged children's data were also included.

## Data

A total of 27,093 women's data, as presented in Fig 1, were extracted from the five rounds of BDHS for analysis (Fig 1). Women were included based on the following criteria: (i) having at least one child within three prior to the survey, (ii) reported delivery methods and place, and (iii) do not have twin or more ordered pregnancy for the most recent live birth.

## Outcome variable

The outcome variable was CS use. The question asked by BDHS to the respondents: "*Was (NAME of the last child occurred in three years) delivered by caesarean, that is, did they cut your belly open to take the baby?*" The response recorded was 'use' (coded as 1) and 'non-use' (coded as 0).

## Exposure variables

The exposure variables considered were the factors at the individual, households, and community level. Maternal age at birth of the last child ($\leq$19, 20–34 years, $\geq$35 years), maternal educational status (no education, primary education, secondary education, higher education), maternal working status (engaged in formal work, not engaged in formal work) were the individual level factors. Another individual level factor considered was antenatal care (ANC) visit classified as no visit, 1–3 visits, $\geq$4 visits. Household level factors were partner's education (no education, primary education, secondary education, higher education), partner's occupation (agricultural worker, physical worker, services, business, other), total children ever born (1–2, >2), wealth status (poorest, poorer, middle, richer, richest), and exposure to mass media (little exposed, moderately exposed, and highly exposed). The BDHS generated the wealth status variable using response on availability of household durable goods, such as radio, refrigerator. Principal component analysis was used for this purpose and the households included were classified in five different levels using the cut off value of 0.20. Moreover, the variable exposure to mass media was generated using women's response on reading newspaper, listening to radio, and watching television at least once a week. Women who reported they did not expose in any of these in the preceding week of the survey were classified as little exposed, exposed one in three in the preceding week of the survey was classified as moderately exposed and

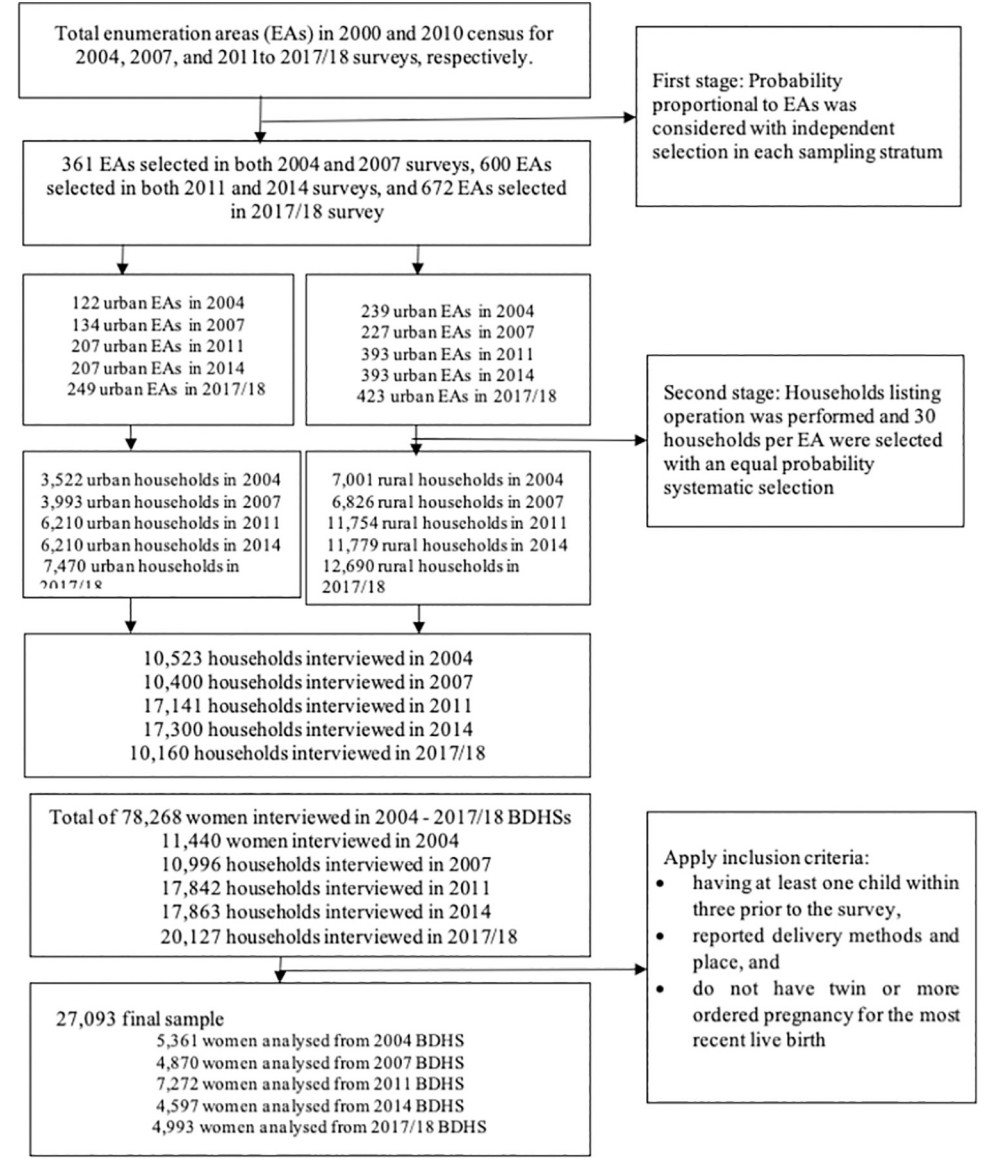

**Fig 1. Flow chart of the study participants selection from the Bangladesh Demographic and Health Surveys, 2004–2017/18.**

exposed at least two in three in the preceding week of the survey were classified as highly exposed. The classification was used in previous studies of Bangladesh [10,27,28]. Finally, place of residence (urban, rural) and region of residence (Barishal, Chattogram, Dhaka, Khulna, Mymensingh, Rajshahi, Rangpur, Sylhet) were considered as community level characteristics.

## Statistical analysis

Descriptive statistics on sampling characteristics and CS use were computed for all surveys. The use of CS across type of health facilities and women's socio-demographic characteristics were also explored. In BDHSs, individuals were nested within a household and households were nested within a cluster. Therefore, the samples are hierarchical. i.e they are not

independent. For such type of hierarchy, multilevel logistic regression model has been recommended in previous research [27–30]. Therefore, multilevel logistic regression models were used in this study to determine the association of CS use with socio-demographic characteristics at five-time points (BDHS surveys: 2004, 2007, 2011 and 2014). Possible pairs of interaction effects were considered in each model. If an interaction effect was found insignificant, the model was deleted and a new model with a different pair was run. Finally, the interactions of the working status and wealth quintile with place of residence produced significant results, therefore, reported in the table. A similar fixed effect model was also run, and its results were compared with the results of multilevel model through using the log-likelihood ratio test. Significance value of this test indicate the suitability of multilevel modelling for this study data. Sampling weights were considered in all analyses. Results are reported as Odds Ratio (OR) and its 95% Confidence Interval (95% CI). The Statistical Package R (version 4.0.3) were used for all statistical analyses. The 'lme4' package in R was used for multilevel modelling. The study was designed and reported following Strengthening the Reporting of Observational Studies in Epidemiology (STROBE) guidelines (S1 Checklist).

## Ethics statement

The BDHSs data collection methods and procedure were reviewed and approved by the institutional review board of ICF and the National Research Ethics Committee of the Bangladesh Medical Research Council. They also provided ethical approval of this survey. Informed consent was obtained from all participants during surveys and data were released in deidentified form. Therefore, no additional ethical approval was required for this study.

## Results

The characteristics of the respondents' analysed are presented in Table 1, separately for each survey. More than two-third of the total women analysed were in the ages of 20–34 years. Around 37% of the women analysed in 2004 survey were illiterate which declined to nearly 6% in 2017 survey. On the contrary, the percentage of higher education increased to nearly 17% in 2017 from only 5% in 2004. Over two-third of the total women did not have any formal working status in all the surveys. The total number of children ever born reported was found to decline from 2004 to 2017. In the 2004 survey, around 53% of the total women had 1–2 children which increased to 72% in the 2017 survey. Percentage of women highly exposed to mass media declined, from 45% in the 2004 survey to 10% in 2017. ANC visit of at least 4 times increased, from 16% in the 2004 survey to nearly 47% in the 2017 survey.

The percentage of institutional delivery, CS use, and changes over the survey years are presented in Table 2. Around half of the total women gave their last birth in healthcare institutions in 2017, which increased from 11% in 2004—a 357% increase in 13 years. At the same time, CS use increased nearly 751%, from 3.88% in 2004 to 33.22% in 2017/18. This increase was nearly 844% for CS use among institutional delivery only, 36.83% in 2004 to 66.66% in 2017/18.

The percentage increase of CS use across several type of health facilities (government, private and non-government) is significantly different (p<0.01, *results not shown*) (Fig 2). In 2004, almost 48% of the CS occurred in government health facilities which declined to nearly 15% in 2017/18. An opposite trend was reported for CS use in private health facilities–increased from 50% in 2004 to 80% in 2017/18—a 226% yearly increase between 2004 and 2017/18.

The CS use across major women's characteristics and place of residence are presented in Fig 3A–3D. A noticeable change in CS use was reported across wealth quintile, education,

**Table 1. Background characteristics of the respondents, BDHS 2004- BDHS 2017/18.**

| | BDHS 2004 (N = 5,361) | BDHS 2007 (N = 4,870) | BDHS 2011 (N = 7,272) | BDHS 2014 (N = 4,597) | BDHS 2017/18 (N = 4,993) |
|---|---|---|---|---|---|
| **Maternal age at birth of the last child** | | | | | |
| ≤19 years | 26.93 (25.51–28.39) | 27.16 (25.61–28.76) | 25.90 (24.73–27.12) | 28.06 (26.39–29.79) | 25.23 (23.86–26.65) |
| 20–34 years | 66.14 (64.6–67.64) | 66.34 (64.73–67.92) | 68.48 (67.20–69.74) | 67.60 (65.80–69.34) | 70.53 (69.05–71.97) |
| ≥35 years | 6.94 (6.23–7.71) | 6.5 (5.75–7.35) | 5.61 (5.01–6.28) | 4.34 (3.65–5.16) | 4.23 (3.68–4.87) |
| **Maternal educational status** | | | | | |
| Illiterate | 36.91 (34.72–39.15) | 26.12 (23.88–28.49) | 19.26 (17.65–20.98) | 14.15 (12.26–16.27) | 6.27 (5.45–7.21) |
| Primary | 30.33 (28.88–31.82) | 30.86 (29.12–32.66) | 30.18 (28.69–31.70) | 27.91 (26.05–29.86) | 27.72 (25.94–29.58) |
| Secondary | 27.44 (25.70–29.26) | 36.36 (34.10–38.67) | 43.17 (41.02–45.25) | 47.71 (45.16–50.28) | 49.02 (47.19–50.85) |
| Higher | 5.33 (4.58–6.18) | 6.66 (5.70–7.78) | 7.39 (6.53–8.36) | 10.23 (9.01–11.59) | 16.99 (15.53–18.56) |
| **Maternal working status** | | | | | |
| Engaged in formal work | 18.47 (16.75–20.32) | 27.67 (25.27–30.20) | 10.18 (9.16–11.30) | 23.68 (21.60–25.89) | 37.38 (35.23–39.57) |
| Not engaged in formal work | 81.53 (79.68–83.25) | 72.33 (69.80–74.73) | 89.82 (88.70–90.84) | 76.32 (74.11–78.40) | 62.62 (35.23–39.57) |
| **Partner's education** | | | | | |
| Illiterate | 39.46 (37.28–41.68) | 34.23 (32.19–36.34) | 28.60 (26.80–30.46) | 23.83 (21.44–26.39) | 13.60 (12.20–15.12) |
| Primary | 27.03 (25.44–28.67) | 28.20 (26.73–29.73) | 29.05 (27.66–30.47) | 29.98 (27.99–32.04) | 22.83 (32.11–35.59) |
| Secondary | 23.76 (22.17–25.43) | 26.29 (24.50–28.16) | 29.35 (27.83–30.91) | 31.79 (29.38–34.30) | 34.17 (32.52–35.85) |
| Higher | 9.75 (8.68–10.94) | 11.27 (10.08–12.59) | 13.01 (11.89–14.21) | 14.41 (13.05–15.87) | 18.41 (16.92–19.99) |
| **Partner's occupation** | | | | | |
| Agriculture worker | 30.64 (28.20–33.19) | 29.44 (26.72–32.30) | 28.08 (26.13–30.13) | 25.66 (22.86–28.66) | 19.60 (17.96–21.36) |
| Physical worker | 41.02 (38.76–43.32) | 43.25 (40.82–45.72) | 42.18 (40.26–44.12) | 44.02 (41.65–46.42) | 53.50 (51.57–55.42) |
| Services | 2.83 (2.37–3.39) | 3.71 (3.13–4.39) | 5.46 (4.73–6.30) | 5.94 (5.02–7.02) | 5.87 (5.11–6.74) |
| Business | 22.58 (20.80–24.46) | 20.80 (19.15–22.55) | 21.86 (20.41–23.39) | 21.63 (19.94–23.41) | 20.81 (19.35–22.34) |
| Other | 2.92 (2.40–3.56) | 2.81 (2.13–3.69) | 2.41 (2.02–2.88) | 2.76 (2.01–3.77) | 0.22 (0.11–0.44) |
| **Total child ever born** | | | | | |
| 1–2 | 53.49 (51.70–55.27) | 58.97 (5.71–60.81) | 63.45 (61.88–64.99) | 70.24 (68.03–72.37) | 71.50 (69.90–73.05) |
| >2 | 46.51 (44.73–48.30) | 41.03 (39.19–42.90) | 36.55 (35.01–38.12) | 29.76 (27.63–31.97) | 28.50 (26.95–30.10) |
| **Exposure to mass media** | | | | | |
| Little exposed | 31.69 (29.30–34.19) | 36.88 (34.26–39.58) | 35.18 (32.92–37.51) | 38.07 (35.15–41.09) | 34.29 (31.87–36.79) |
| Moderately exposed | 22.94 (21.67–24.26) | 22.50 (21.11–23.97) | 48.70 (46.64–50.76) | 48.15 (45.56–50.75) | 55.23 (52.94–57.51) |
| Highly exposed | 45.36 (43.02–47.73) | 40.62 (38.22–43.06) | 16.12 (14.94–17.37) | 13.78 (12.40–15.28) | 10.47 (9.40–11.65) |
| **Antenatal care use** | | | | | |
| No use | 44.22 (41.46–47.02) | 39.90 (37.10–42.77) | 35.50 (33.37–37.69) | 21.45 (18.63–24.58) | 8.03 (7.00–9.19) |
| 1–3 visits | 39.87 (37.89–41.88) | 39.50 (37.46–41.57) | 40.70 (38.98–42.44) | 47.38 (45.10–49.67) | 45.10 (43.11–47.11) |
| ≥ 4 visits | 15.90 (14.27–17.69) | 20.60 (18.47–22.91) | 23.80 (22.20–25.49) | 31.17 (28.60–33.86) | 46.87 (44.64–49.12) |
| **Household wealth status** | | | | | |
| Poorest | 23.99 (21.86–26.25) | 21.79 (19.49–24.27) | 22.05 (10.15–24.08) | 21.62 (18.97–24.53) | 20.66 (18.62–22.86) |
| Poorer | 20.73 (19.27–22.27) | 21.38 (19.77–23.08) | 19.97 (18.77–21.23) | 18.91 (17.24–20.71) | 20.51 (19.00–22.12) |
| Middle | 19.58 (18.18–21.07) | 18.97 (17.26–20.81) | 19.83 (18.54–21.18) | 19.12 (17.16–21.25) | 19.28 (17.75–20.91) |
| Richer | 18.43 (16.79–20.19) | 19.60 (17.69–21.66) | 19.69 (18.20–21.27) | 20.66 (18.60–22.89) | 20.16 (18.42–22.02) |
| Richest | 17.27 (15.36–19.36) | 18.26 (16.32–20.39) | 18.46 (16.97–20.05) | 19.69 (17.22–22.42) | 19.39 (17.48–21.44) |
| **Place of residence** | | | | | |
| Urban | 20.59 (18.99–22.29) | 21.12 (19.79–22.52) | 23.29 (22.10–24.53) | 26.11 (23.49–28.92) | 26.76 (25.07–28.53) |
| Rural | 79.41 (77.71–81.01) | 78.88 (77.48–80.21) | 76.71 (75.47–77.90) | 73.89 (71.08–76.51) | 73.24 (71.47–74.93) |
| **Region of residence** | | | | | |
| Barishal | 6.18 (5.30–7.20) | 6.38 (5.82–6.99) | 5.83 (5.28–6.32) | 5.80 (4.83–6.95) | 5.72 (5.15–6.35) |
| Chattogram | 20.55 (19.27–21.89) | 20.95 (19.22–22.78) | 21.64 (20.48–22.84) | 21.75 (19.32–24.39) | 21.08 (19.42–22.84) |

*(Continued)*

**Table 1.** (Continued)

| | BDHS 2004 (N = 5,361) | BDHS 2007 (N = 4,870) | BDHS 2011 (N = 7,272) | BDHS 2014 (N = 4,597) | BDHS 2017/18 (N = 4,993) |
|---|---|---|---|---|---|
| Dhaka | 30.90 (29.03–32.83) | 31.74 (29.84–33.70) | 31.46 (30.05–32.90) | 35.45 (31.25–39.88) | 25.62 (23.89–27.44) |
| Khulna | 11.19 (10.28–12.17) | 10.22 (9.26–11.28) | 9.69 (9.06–10.37) | 8.05 (7.14–9.06) | 9.22 (8.37–10.16) |
| Mymensingh | | | | | 8.54 (7.68–9.49) |
| Rajshahi | 23.82 (22.06–25.67) | 22.88 (20.96–24.92) | 13.52 (12.40–14.73) | 10.02 (8.84–11.35) | 11.59 (10.35–12.97) |
| Rangpur | | | 10.96 (10.26–11.71) | 9.73 (8.17–11.56) | 10.64 (9.59–11.80) |
| Sylhet | 7.36 (6.46–8.38) | 7.83 (6.43–9.50) | 6.90 (6.45–7.37) | 9.20 (7.03–11.96) | 7.57 (6.71–8.53) |

working status, and place of residence. In 2004, CS use was mostly prevalent among richest with nearly 70% of the total use where the prevalence of CS use among poorest was <1%. However, over time, CS use increased in all wealth quintiles, though, predominantly higher among women with middle to richest wealth quintile with over 78% of the total CS use. The poorest and poorer groups, which represent around one third of the total population with very high fertility rate, utilised only 22% of the total CS in Bangladesh. Similarly, in all the years, women with secondary and higher education jointly employed over 80% of the total CS use whereas remaining 20% CS use was found among illiterate and primary educated women. In 2004, over 61% of the total CS use reported by the women who resided in urban area which declined to 36% in 2017/18. An opposite trend was reported for women living in rural areas, a increase from 39% in 2004 to 64% in 2017/18.

We further explored three characteristics CS use among three characteristics (maternal education, household wealth quintile and place of residence) (Fig 4). Around 57% of the women who had secondary or above education, in the middle to higher wealth quintile and resided in urban area were found to use CS. However, women who had lower education, lower quintile and resided in rural area, reported only 12% CS use in 2017, which is an increase of only 0.27% in 2004. Around 24% of the mothers who had at least two disadvantages' characteristics among three (lower education, lower wealth quintile and rurality) reported CS use in 2017, increased from nearly 3% in 2004.

**Table 2. Institutional delivery and caesarean delivery rates and changes in Bangladesh, 2004 to 2017/18.**

| Survey | Period | Number of births | Delivery rate, % (95% CI) | | | Percentage change across the years | | |
|---|---|---|---|---|---|---|---|---|
| | | | Institutional[1] | Population caesarean[2] | Institutional caesarean[3] | Institutional | Population caesarean | Institutional caesarean |
| BDHS 2004 | 2001–2003 | 5361 | 10.89 (9.63–12.30) | 3.88 (3.25–4.62) | 36.13 (31.66–40.85) | – | – | – |
| BDHS 2007 | 2004–2006 | 4870 | 16.36 (14.65–18.24) | 8.51 (7.41–9.76) | 52.54 (47.99–57.06) | 50.22 (48.29–53.98) | 119.33 (111.26–128.00) | 454.19 (396.82–515.79) |
| BDHS 2011 | 2006–2010 | 7272 | 26.51 (24.87–28.22) | 14.90 (13.74–16.14) | 56.89 (54.09–59.65) | 143.43 (129.43–163.33) | 284.02 (249.35–322.77) | 574.59 (460.22–708.46) |
| BDHS 2014 | 2010–2013 | 4597 | 38.69 (35.86–41.60) | 24.18 (22.11–26.37) | 62.64 (59.77–65.42) | 255.28 (238.21–282.20) | 523.20 (470.77–580.30) | 733.74 (604.65–887.87) |
| BDHS 2017 | 2014–2016 | 4993 | 49.89 (47.60–52.19) | 33.02 (31.09–35.00) | 66.61 (64.27–68.88) | 357.20 (324.31–394.28) | 751.03 (657.57–856.62) | 843.62 (686.17–1003.0) |

Note

[1]Percentage of total delivery occurred in any forms of healthcare facilities

[2]Percentage of total cesarean delivery without considering delivery places

[3]Percentage of total cesarean delivery among institutional delivery.

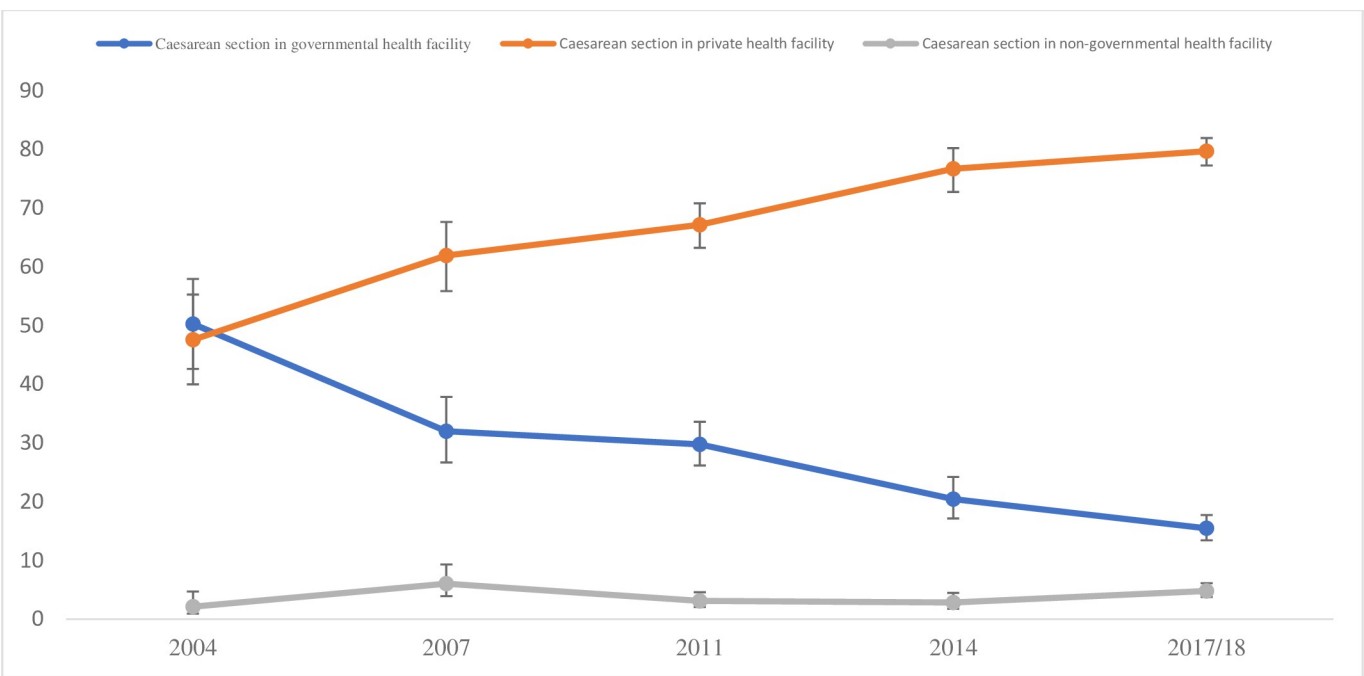

**Fig 2. Caesarean section use across type of health facility, Bangladesh 2004-2017/18.**

The results of the multilevel logistic regression model to determine the factors associated with CS use are presented in Table 3. Each model results were compared with their corresponding fixed effect model results though log-likelihood test. The test results were significant for all surveys, which indicate use of multilevel model was appropriate. The cluster level variations of CS use was reported higher for the 2004 survey at around 27%, which was found gradually decreased with the recent surveys and around 11% variability of CS use was reported for the 20-17/18 survey. Overall, the risk factors of CS use were different among the survey years and different sub-group of socio-demographic characteristics. Increased age, higher education, higher partner education, increase number of ANC visit, and improved socio-economic status of the respondent were associated with a higher likelihood of CS use in all the survey years. Women who had more than 2 children and resided in Barishal division were found to have lower likelihood of CS use.

From the interaction terms, women's rural residence and their engagement in formal job were associated with reduced likelihood of CS use in all the surveys. In 2004, this interaction was found to be associated with 11% (OR, 0.89, 95% CI, 0.71–0.99) reduction of CS use, which was further increased to 51% (OR, 0.49, 0.03–0.65) reduction of CS use in 2017/18. Similarly, consistent lower odds of CS use were found among rural poorest and poorer women with a rising trend of declining over the years. In 2004, 12% (OR, 0.45–0.98) declined likelihood of CS use was found among rural poorest women. This likelihood was 66% (OR,44, 95% CI, 0.22–0.88) lower in the 2017/18 survey. Comparably, around 10% (OR, 0.90, 95% CI, 0.79–0.99) lower likelihood of CS use was found among rural poorer women in 2004 survey, which increased to 48% (OR, 0.52, 95% CI, 0.32–0.89) declined in the 2017/18 survey.

## Discussion

Nearly one-third of the total women in Bangladesh use CS as per the 2017/18 survey which is around 751% higher than the 3.88% CS use in the 2004 survey. Private health facilities had

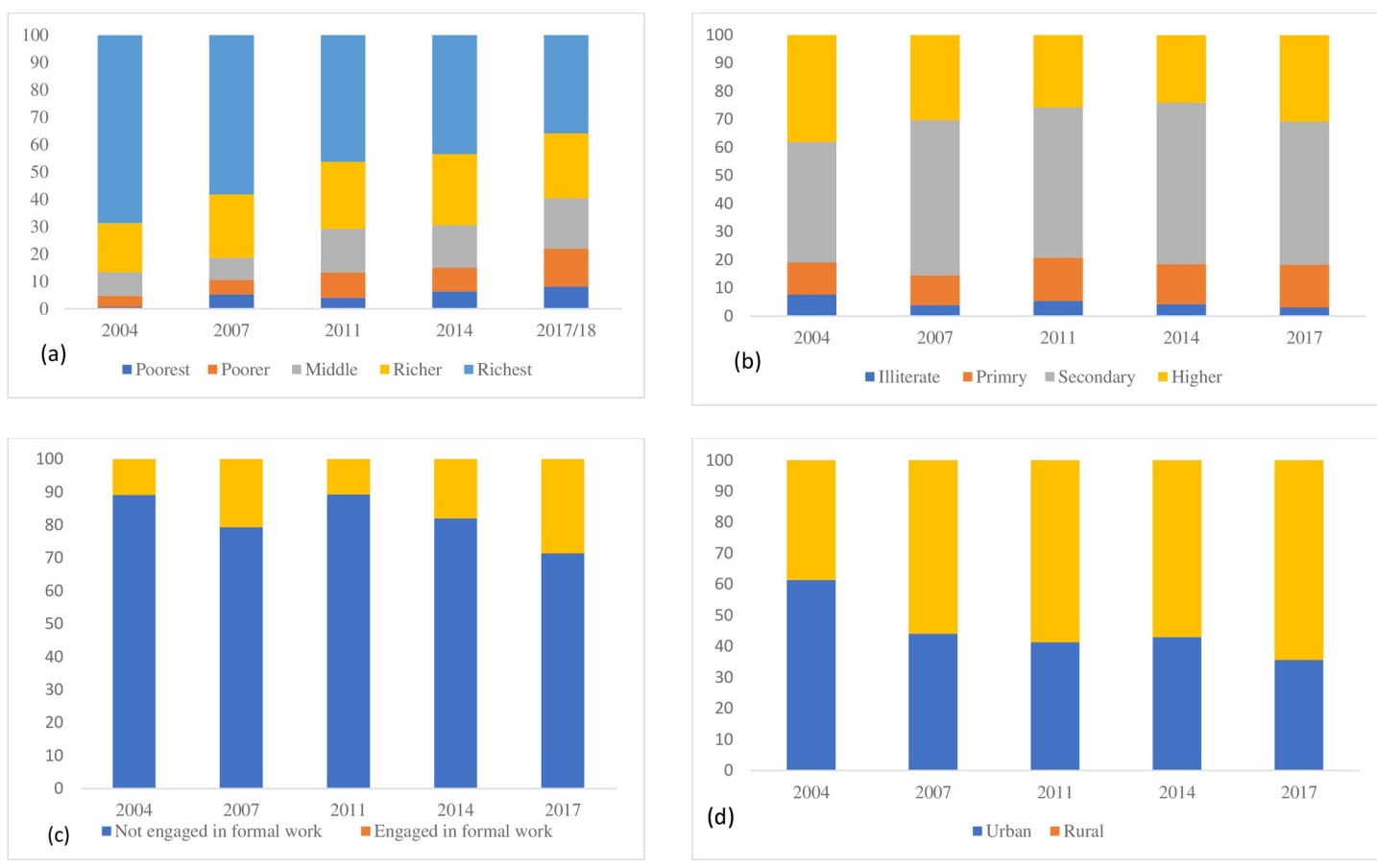

**Fig 3. (a-d)** Distribution of Socioeconomic Status among women having caesarean section use in Bangladesh, 2004-2017/18.

became popular in providing the CS delivery over the survey years and now providing nearly 80% of the total CS in Bangladesh. Alternatively, the share of the government health facilities in providing CS declined substantially, from 50.29% in 2004 to 15.47% in 2017/18. Private health facilities in Bangladesh are mostly profit ambitious and located in urban centre [11]. Consequently, respondents living in rural area, that cover over 70% of the total population in Bangladesh, with poor economic condition could not access this service—as this study reports. This indicates a group of women in Bangladesh use CS abusively while other group could not access this service even under necessary condition. As a result, they are more likely to face short-term and long-term health consequences, problem in the subsequent pregnancies, as well as death from obstetric complications. Efforts are needed to prevent medically unnecessary CS use along with guaranteed access to save delivery for rural and poor women.

This study reported a rapid rise on CS use in Bangladesh in general and in private health facilities in particular. The use of CS declined substantially in the government health facilities. This trend contradicts to what was reported worldwide and in LMICs [31], including India [32], Pakistan [33] and Ethiopia [34]. Though such exponential rising rate of CS use in private health facilities in Bangladesh has two different roots, both from the individual and health facilities level, however, the government poor regulation on the private health facilities is the major factor.

Over the decade, CS use has been rising exponentially and becoming popular in Bangladesh, mainly because of its increasing availability in the health facilities on women's demand

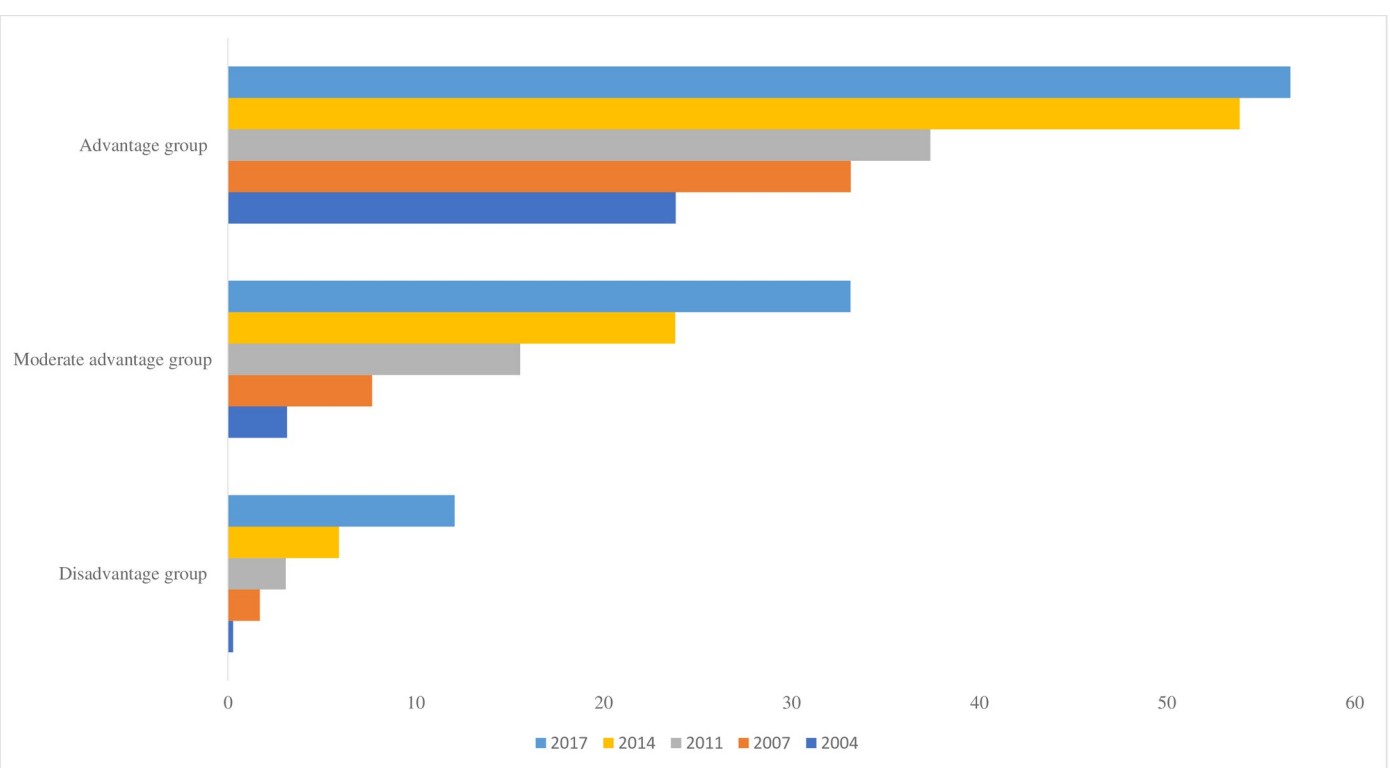

Note: Advantage group: higher education, higher wealth quintile and urban place of residence; moderate advantage group: At least one advantage charecteristics among three: higher education, higher wealth quintile and urban place of residence, moderate advantage group; disadvantage group: lower education, lower wealth quintile and rural place of residence

**Fig 4. Change in caesarean section use across type of disadvantages characteristics in Bangladesh, 2004-2017/18.**

without medical necessity [35]. Women are also misguided by the healthcare personnel and their peers who had undergone CS with wrong insight over this method [11]. They often consider CS use as a way of reducing labor pain and safe approach for both women and upcoming newborn, even where there were no medical necessity [22,36]. This is the primary reason of booming the CS use in the private health facilities as CS use on women's demand is mostly available there–contradict to the government health facilities practice [19]. Moreover, a large percentage of women admitted in the government health centre, finally changes their mind and shift to the private facilities and choose to undergo CS [37,38]. The reasons are lack of healthcare personnel in the governmental health facilities, poor quality of care, long waiting time for the doctors and unsupportive behaviours of the nurses and other associated medical staff [19,38]. Conversely, in the private health facilities, even though they are profit making institutions, they always ensure better care and concern for the patients [19]. The brokers are also active in the governmental health facilities independently and/or in cooperation with the corrupt healthcare personnel to motivate women to shift to the private health facilities by promising the advantages and better care provided by the CS [39]. About 15% use of CS in the government health facilities are mainly because of their main role on handling more complicated cases who are directly admitted there as well as referred from the private and non-governmental health facilities [19].

It is found in this study that such rising rate of CS as well as increase familiarization of private facilities for CS contribute to double burden of CS in Bangladesh. A similar trend is found in other LMICs; however, the strength is much lower than Bangladesh [40–42]. The average

**Table 3. Multilevel logistic regression models for the association of caesarean delivery with socio-demographic characteristics and the interactions of the working status and wealth quintile with place of residence at five-time points (BDHS Surveys: 2004, 2007, 2011, 2014, 2017/18).**

| Characteristics | BDHS 2004, OR (95% CI) | BDHS 2007, OR (95% CI) | BDHS 2011, OR (95% CI) | BDHS 2014, OR (95% CI) | BDHS 2017/18, OR (95% CI) |
|---|---|---|---|---|---|
| **Maternal age at birth of the last child** | | | | | |
| ≤19 (ref) | 1.00 | 1.00 | 1.00 | 1.00 | 1.00 |
| 20–34 | 1.45 (0.97–2.16) | 1.53 (1.10–2.12)*** | 1.34 (1.09–1.66)*** | 1.28 (1.021.62)** | 1.35 (1.11–1.64)*** |
| ≥35 | 3.77 (1.65–8.63)*** | 2.87 (1.34–6.13)*** | 1.92 (1.13–3.26)*** | 2.18 (1.27–3.47)*** | 1.93 (1.28–2.91)*** |
| **Maternal educational status** | | | | | |
| Primary (ref) | 1.00 | 1.00 | 1.00 | 1.00 | 1.00 |
| No education | 1.18 (0.61–2.28) | 0.71 (0.35–1.43) | 0.98 (0.65–1.45) | 0.80 (0.52–1.23) | 1.18 (0.78–1.77) |
| Secondary | 1.43 (0.87–2.33) | 2.29 (1.47–3.58)*** | 1.20 (0.95–1.51) | 1.46 (1.06–2.03)** | 1.48 (1.19–1.84)*** |
| Higher | 2.28 (1.14–4.56)** | 2.86 (1.70–4.79)*** | 2.05 (1.44–2.92)*** | 1.73 (1.16–2.57)** | 2.04 (1.50–2.78)*** |
| **Women's working status** | | | | | |
| Engaged in formal work (ref) | 1.00 | 1.00 | 1.00 | 1.00 | 1.00 |
| Not engaged in formal work | 1.18 (0.72–1.93) | 1.25 (0.83–1.89) | 1.78 (1.25–2.53)*** | 1.56 (1.00–2.41)** | 1.07 (0.79–1.43) |
| **Partner's educational status** | | | | | |
| Primary (ref) | 1.00 | 1.00 | 1.00 | 1.00 | 1.00 |
| No education | 0.64 (0.34–1.21) | 1.26 (0.73–2.16) | 0.72 (0.51–1.01) | 0.85 (0.52–1.37) | 1.04 (0.78–1.39) |
| Secondary | 1.06 (0.63–1.79) | 1.43 (0.97–2.10) | 1.11 (0.89–1.40) | 1.23 (0.93–1.63) | 1.16 (0.94–1.42) |
| Higher | 1.65 (0.90–3.02) | 1.82 (1.18–2.82)*** | 1.76 (1.29–2.41)*** | 1.73 (1.19–2.50)*** | 1.53 (1.14–2.06)*** |
| **Partner's occupation status** | | | | | |
| Agricultural worker (ref) | 1.00 | 1.00 | 1.00 | 1.00 | 1.00 |
| Physical worker | 1.24 (0.65–2.38) | 1.78 (1.02–3.12)** | 1.16 (0.90–1.49) | 1.07 (0.76–1.50) | 1.05 (0.83–1.34) |
| Services | 1.83 (0.87–3.88) | 2.22 (1.20–4.12)*** | 1.32 (0.92–1.88) | 1.83 (1.13–2.97)*** | 1.54 (1.02–2.32)** |
| Business | 1.07 (0.55–2.07) | 2.71 (1.61–4.54)*** | 1.20 (0.93–1.54) | 1.61 (1.16–2.22)*** | 1.08 (0.83–1.42) |
| Other | 2.28 (0.87–5.91) | 2.13 (1.01–4.50)*** | 0.64 (0.38–1.09) | 1.66 (0.94–2.92) | 0.93 (0.24–3.54) |
| **Total children ever born** | | | | | |
| 1–2 (ref) | 1.00 | 1.00 | 1.00 | 1.00 | 1.00 |
| >2 | 0.38 (0.24–0.60)*** | 0.44 (0.29–0.66)*** | 0.57 (0.45–0.72)*** | 0.62 (0.46–0.82)*** | 0.64 (0.52–0.78) |
| **Exposure to mass media** | | | | | |
| Little exposed | 1.00 | 1.00 | 1.00 | 1.00 | 1.00 |
| Moderately exposed | 0.83 (0.37–1.88) | 0.82 (0.48–1.1.39) | 1.48 (1.14–1.93)*** | 0.97 (0.73–1.26) | 1.54 (1.26–1.89)*** |
| Highly exposed | 1.11 (0.51–2.42) | 1.03 (0.60–1.75) | 4.02 (3.02–5.35)*** | 1.09 (0.76–1.57) | 1.46 (1.06–2.02)** |
| **Antenatal care use** | | | | | |
| No use | 1.00 | 1.00 | 1.00 | 1.00 | 1.00 |
| 1–3 visits | 4.40 (1.90–10.19)*** | 3.00 (1.69–5.32)*** | 2.32 (1.76–3.06)*** | 3.02 (1.91–4.78)*** | 3.06 (1.62–5.78)*** |
| ≥ 4 visits | 12.29 (5.15–29.33)*** | 8.47 (4.82–14.88)*** | 4.02 (3.02–5.35)*** | 4.79 (2.99–7.69)*** | 5.83 (3.10–10.98)*** |
| **Household wealth status** | | | | | |
| Middle (ref) | 1.00 | 1.00 | 1.00 | 1.00 | 1.00 |
| Poorest | 0.25 (0.03–1.99) | 1.98 (0.31–12.63) | 2.13 (0.87–5.20) | 0.26 (0.11–0.62)*** | 0.44 (0.23–0.83)** |
| Poorer | 0.67 (0.15–2.90) | 1.62 (0.50–5.17) | 2.42 (0.98–5.99) | 1.36 (0.61–3.05) | 0.73 (0.36–1.47) |
| Richer | 0.51 (0.14–1.93) | 2.79 (0.95–8.22) | 2.09 (1.03–4.24)** | 1.10 (0.62–1.95) | 0.87 (0.57–1.33) |
| Richest | 1.44 (0.57–3.67) | 4.94 (2.00–12.19)*** | 4.08 (2.28–7.32)*** | 1.81 (1.04–3.14)** | 1.44 (0.89–2.31) |
| **Place of residence** | | | | | |
| Urban (ref) | 1.00 | 1.00 | 1.00 | 1.00 | 1.00 |
| Rural | 0.17 (0.04–0.75) | 2.62 (0.88–7.87) | 4.04 (1.84–8.89)*** | 0.78 (0.35–1.73) | 0.62 (0.37–1.02) |
| **Region of residence** | | | | | |
| Dhaka (ref) | 1.00 | 1.00 | 1.00 | 1.00 | 1.00 |

*(Continued)*

**Table 3.** (Continued)

| Characteristics | BDHS 2004, OR (95% CI) | BDHS 2007, OR (95% CI) | BDHS 2011, OR (95% CI) | BDHS 2014, OR (95% CI) | BDHS 2017/18, OR (95% CI) |
|---|---|---|---|---|---|
| Barishal | 0.61 (0.32–1.14) | 0.44 (0.24–0.78)*** | 0.66 (0.46–0.95)** | 0.64 (0.46–0.88)*** | 0.71 (0.52–0.98)** |
| Chattogram | 0.44 (0.27–0.73) | 0.47 (0.31–70.41)*** | 0.71 (0.55–0.97)** | 0.51 (0.38–0.69)*** | 0.54 (0.40–0.73)*** |
| Khulna | 0.79 (0.48–1.28) | 0.67 (0.43–1.02) | 1.49 (1.11–1.99)*** | 1.28 (0.95–1.72) | 1.25 (0.91–1.70) |
| Mymensingh | | | | | 0.77 (0.56–1.04) |
| Rajshahi | 0.64 (0.36–1.10) | 0.55 (0.36–0.84)*** | 1.02 (0.76–1.38) | 0.92 (0.67–1.26) | 1.00 (0.73–1.37) |
| Rangpur | | | 0.66 (0.48–0.92)** | 0.58 (0.38–0.91)** | 0.78 (0.55–1.09) |
| Sylhet | 0.99 (0.58–1.71) | 0.57 (0.38–0.86)*** | 1.08 (0.79–1.48) | 0.48 (0.34–0.66)*** | 0.62 (0.46–0.82)*** |
| **Interaction of working status and place of residence** | | | | | |
| no#rural | 89 (0.71–99)** | 0.76 (0.40–0.93)*** | 0.51 (0.29–0.88)*** | 0.83 (0.49–0.99)*** | 0.49 (0.03–0.65)*** |
| **Interaction of wealth quintile and place of residence** | | | | | |
| poorest#rural | 0.88 (0.45–0.98)*** | 0.68 (0.29–0.89)*** | 0.17 (0.07–0.46)*** | 0.67 (0.52–0.74)*** | 0.44 (0.22–0.88)*** |
| poorer#rural | 0.90 (0.79–0.99)*** | 0.52 (0.13–0.86)*** | 0.32 (0.12–0.82)*** | 0.48 (0.19–0.79)*** | 0.52 (0.32–0.89)*** |
| richer#rural | 3.54 (0.77–16.32) | 0.52 (0.16–1.76) | 0.58 (0.27–1.24) | 1.31 (0.63–2.68) | 1.38 (0.83–2.30) |
| richest#rural | 1.23 (0.38–3.94) | 0.59 (0.21–1.68) | 0.37 (0.19–0.71)*** | 1.31 (0.65–2.63) | 1.47 (0.84–2.58) |
| **Random effects (measure of variation for caesarean delivery)** | | | | | |
| Cluster-level variance | 0.65 (0.04)*** | 0.59 (0.07)*** | 0.45 (0.04)*** | 0.31 (0.05)*** | 0.26 (0.06)*** |
| Intra-class correlation (ICC) | 27.04% | 24.36% | 19.12% | 15.26% | 11.08% |
| Log-likelihood for fixed effects to random effects model | 330.78*** | 325.28*** | 298.75*** | 347.15*** | 370.78*** |

Note

**p<0.05

***p<0.001.

cost of performing CS in the private health facilities in Bangladesh is 612 USD, twice higher than the average monthly income (USD 301) of Bangladeshi population and 6 to 8 fold higher than the Bangladeshi poorer population [6,43]. For rural women, such higher cost is accompanied with transportation and relocation costs to urban area where all private health facilities and majority of the government health facilities are located. Therefore, poorer women in general and rural poor women in particular could not use this service. These increase the rate of home delivery, particularly among the rural women who usually have higher number of children [10,28]. In this case, most women depend on their experience from the previous pregnancies rather than accessing services including ANC, which is the motivator of delivery care access. Our findings are supportive to this conclusion, and it is similar to what has been reported in other LMICs [33,37,40,42]. Even if the use of CS is required, women living in rural area, particularly those who have no proper income and belong to the poor socio-economic condition, often need to sell their properties, or borrow money from the usurer [35]. The findings of the interaction effects of place of residence with women of no formal job engagement and poor wealth quintile have been justified through these linkages. Such burden is even intensified because of the government single-mindedness to reduce the number of CS rather than considering the justification of its importance and requirements, particularly among the disadvantage groups. However, this issue may not be true for the urban poor women as the non-governmental health facilities there provide free services or with a very minimum cost [12].

The government policy and programs to control the CS use are also not properly effective. Instead of decreasing, it often leads to an increase or double the burden of CS use. In 2019 a committee of experts and stakeholders was set up following the High Court (one of the two wings of the supreme court of Bangladesh) order to stop unnecessary CS use [35]. The committee later introduced a system whereby all health facilities were to report history of all deliveries carried out by filling a form, either normal or CS. For CS, the reasons of choosing and conducting CS also need to be reported. These procedures contradict to what is recommended by the International Federation of Gynaecology and Obstetrics for control CS: imposing the fixed fee for live birth, obliging hospital to publish their statistics and fully informing women about the risks of CS use [44]. The government also does not impose any restriction on private facilities to provide delivery care along with CS as part of corporate social responsibility or based on mutual agreement with the government where the government will support for CS for the disadvantages' groups. No programs are also carried out to provide delivery services along with CS requirement at the rural areas. For instance, the government does not take any policies and programs to ensure CS at the community clinics or union health complex, though they are higher in number, located in rural areas for every 6000 population and currently playing a major role to provide ANC and postnatal care [30]. Consequently, the current programs to control CS does not work properly in either way, controlling CS as well as reducing the double burden of CS. These suggest urgent need for the policies and programs from the country level to effectively regulate the private health facilities on CS use as well as ensuring accessing to CS for those who do not afford or have the proper financial support.

This study has a number of strengths and limitations. The major limitation is the analysis of cross-sectional data; therefore, findings are correlational and not casual. Moreover, the data were collected retrospectively which indicate a possibility of recall error, particularly for the variable like ANC visits. In addition, since the data on the CS medical requirement is not available, our explanations about the double burden are based on probability rather than considering whether such use is medically justified. However, these interaction effects will enable policy makers to identify the group of women who could not access this service and develop policies and programs accordingly. Furthermore, other than the factors adjusted in the model, there are many other factors including transportation and quality of the care which are also important determinant of CS use. We could not adjust them in the model because of lack of data. Regardless of these limitations, this study is the first of its kind in Bangladesh and other LMICs that bring the issue of double burden of CS into focus. The factors associated with the CS along with the interaction effects of the maternal disadvantage characteristics are also determined. This will enable the policy makers to plan for more effective policies and programs.

## Conclusion

This study found a rapid rise in CS use over time, from nearly 4% in 2004 to 33% in 2017/18. Nearly 80% of the CS are performed in the private health facilities, increased from about 48% in 2004. In contrary, CS use declined from nearly 50% in 2004 to only 15% in 2017/18 in the governmental health facilities. Such rising of CS use is mainly due to availability of CS in the private health facilities which could also lead to a double burden of CS in Bangladesh. Since private health facilities are profit ambitious, women with improved socio-economic condition are the main clients to this service. On the other hand, rural, not engage in formal work, and poorer mothers could not afford this service. This goes along with the government health facilities level challenges including lack of healthcare personnel and poor quality of care, restrict disadvantages women in accessing CS even when it is for a medical requirement. This

indicates a pathway to increase short term and long-term obstetric and medical complications as well as deaths from obstetric complications. Therefore, government programs and regulations are strictly required in order to determine the need for CS as well as to control CS use in general and private health facilities in particular. The government should also impose regulations on the private health facilities to deliver CS when required medically among disadvantages with minimum fees or even free of charge when necessary.

## Supporting information

**S1 Checklist. STROBE statement.**
(DOCX)

## Acknowledgments

Special thanks to the MEASURE DHS from the authors for granting access to the 2017/18 BDHS data.

## Author Contributions

**Conceptualization:** Md Nuruzzaman Khan.

**Data curation:** Md Nuruzzaman Khan.

**Formal analysis:** Md Nuruzzaman Khan.

**Investigation:** Md Nuruzzaman Khan.

**Methodology:** Md Nuruzzaman Khan.

**Project administration:** Md Nuruzzaman Khan.

**Software:** Md Nuruzzaman Khan.

**Supervision:** Md Mostafizur Rahman.

**Visualization:** Md Nuruzzaman Khan.

**Writing – original draft:** Md Nuruzzaman Khan.

**Writing – review & editing:** Md Awal Kabir, Asma Ahmad Shariff, Md Mostafizur Rahman.

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
