## [Decision Letter · Decision Letter 0]

26 Oct 2021

PGPH-D-21-00594

Too many yet too few caesarean section deliveries in Bangladesh: an ongoing public health challenge to improve maternal and child health

Dear Dr. Khan,

Thank you for submitting your manuscript to PLOS Global Public Health. After careful consideration, we feel that it has merit but does not fully meet PLOS Global Public Health’s publication criteria as it currently stands. Therefore, we invite you to submit a revised version of the manuscript that addresses the points raised during the review process.

We look forward to receiving your revised manuscript.

Kind regards,

Javier H Eslava-Schmalbach, M.D., Ph.D., MSc

Academic Editor

Journal Requirements:

1. Please provide separate figure files in .tif or .eps format only, and remove any figures embedded in your manuscript file.  If you are using LaTeX, you do not need to remove embedded figures.

2. Tables should not be uploaded as individual/separate files.  Please remove these files and include the tables in your manuscript file.

3. Since your data is not available for proprietary reasons, please explain via email why the data is not available. Please also include the contact information for the third party organization that should be contacted should other researchers want to request access to this data and please include the full citation of where the data can be found. We also request that you verify with us via email that any researcher will be able to obtain the data set in the same manner that the you have obtained it. If you feel you are unwilling or unable to adhere to this policy, please explain your reasons by return email and your exemption request will be escalated to the editor for approval. Your exemption request will be handled independently and will not hold up the peer review process, but will need to be resolved should your manuscript be accepted for publication. One of the Editorial team will be in touch if they require more information.

Additional Editor Comments (if provided):

Dear Authors: Please answer/include each one of the reviewers' suggestions. Also it is needed that you review the following aspects:

1. The title does not include the design, as Strobe suggests.

2. The sample size of each one of the rounds should be included in the methods, and each one of the tables in which they are used.

3. The adequate use of sampling weights when the analysis was performed, should be included in the methods section

4. Given that the original sample size was not estimated to find prevalence at facility and division levels, a paragraph should be included to explain how authors handled this limitation for the analysis and for the discussion of their results.

5. In figure 1, confidence intervals of values should be included in the figure

6. In Table 2. a definition of "institutional", "population caesarean" and "institutional caesarean", should be included as a table-note

7. In Table 3, Interaction of working status and place of residence values for 2004 should be double checked

8. In Table 3 add table-notes to make it clearer

9. In Figure 3 y-labels used are not clear. It is not clear how are they different. If rural place and lower wealth quintile are common among them, these characteristics could be included in the title of the figure. The y-label should use the additional disadvantage that is different. Please use understandable labels (the label of the figure 3b label is not understandable)

10. Ethics statement should be added to the manuscript

Reviewers' comments:

Reviewer's Responses to Questions

**Comments to the Author**

1. Does this manuscript meet PLOS Global Public Health’s publication criteria? Is the manuscript technically sound, and do the data support the conclusions? The manuscript must describe methodologically and ethically rigorous research with conclusions that are appropriately drawn based on the data presented.

Reviewer #1: Partly

Reviewer #2: Partly

Reviewer #3: Yes

2. Has the statistical analysis been performed appropriately and rigorously?

Reviewer #1: Yes

Reviewer #2: I don't know

Reviewer #3: Yes

3. Have the authors made all data underlying the findings in their manuscript fully available (please refer to the Data Availability Statement at the start of the manuscript PDF file)?

Reviewer #1: Yes

Reviewer #2: Yes

Reviewer #3: Yes

4. Is the manuscript presented in an intelligible fashion and written in standard English?

Reviewer #1: Yes

Reviewer #2: No

Reviewer #3: Yes

5. Review Comments to the Author

Reviewer #1: The article “Too many yet too few caesarean section deliveries in Bangladesh: an ongoing public

health challenge to improve maternal and child health” is a good effort from the authors to address a very important health issue of the developing world. The authors should address following suggestions to improve their manuscript,

Abstract: it is too lengthy, especially the background section, kindly reduce the number of words so that readers can have a better understanding.

Introduction

Kindly keep the format of citations uniform (space between the citation and the last word). “Only 10% of the total countries worldwide have the CS rate 10-15%[3],”

Kindly cite articles from the developing countries that are on the same topic (A Cross-Sectional Study to Assess the Frequency and Risk Factors Associated with Cesarean Section in Southern Punjab, Pakistan. Int. J. Environ. Res. Public Health 2021, 18, 8812. https://doi.org/10.3390/ijerph18168812).

Methods

Was there any ethical approval taken for the study? Or it was not required? Kindly mention it clearly.

Results

Table 1, Kindly add in the method section how exposure to the mass media was categorized?

How were the participants categorized into different wealth status?

Figure 2: kindly provide y-axis on every subset separately as it is hard to follow in its current form.

Figure 3: the title is hard to follow kindly rephrase it to make it easy for the readers.

Reviewer #2: The article is very interesting and addresses a timely issue. It addresses the challenges of maternal mortality by addressing the issue of inequity of care in the delivery of maternal services.

Title

The study design must be specified in the title

Key words

The keywords must be reviewed : choose between double burden and triple burden : using both are confusing.

Introduction

Bangladesh is one of the LMICs, so it is unnecessary to single it out in paragraph 4 of the introduction. The reference to anemia and nutritional problems at this point is not justified as it is not relevant to the topic. The authors should review the relevance of this sentence in the introduction. The authors specify that there were differences in the use of CS in certain geographic regions. Can they show the specifics of the regions in question?

The objectives of the study were clearly stated as well as its relevance.

Methods

The "exposure variables" section should be better described. It would be better to have a comprehensive description of the exposure variables and their nature.

Can the authors indicate whether the sampling weight was used in the descriptive statistics to adjust for the non-proportional distribution of the sample across different regions and their urban and rural areas, as suggested by the DHS procedure

- The use of multilevel modeling is relevant. They specified it by the hierarchical structure of the data. However, they need to make explicit in the variable exposure part, which variables are individual level and which are community level.

- Tell the version of the R software that was used and specify the package that was used to do the multilevel modeling.

- It is assumed that the yearly models have been fitted on the individual and community level variables. However, we have no idea of the variability of the clusters. To do this, we need to give the intra-class correlation coefficient.

- Also, the authors need to be more consistent in the presentation of Table 3. Usually the references are the least exposed. Authors choose certain references like middle over other wealth modalities. The notion of reference mode also applies to qualitative explanatory variables. In fact, in a model, all the coefficients are calculated in relation to the reference mode. It is important to choose a reference mode that makes sense in order to facilitate interpretation. Moreover, this choice may also depend on the way in which one wishes to present the results. In general, one should avoid choosing as a reference an item that is poorly represented in the sample or an item that corresponds to an atypical situation. Can the authors specify the reasons for the choice of reference modalities?

Add the diagram of study flow in the article to help understand the sampling

Figure 2: The authors list region of residence among the socioeconomic factors. Unless we describe the regions and highlight their socioeconomic differences, we suggest removing them from this figure

Figure 2: (c) the part "engaged in formal worked/not engaged in formal worked" is not part of the authors' analysis models and therefore should not appear in this work. On the other hand, the status (worked: yes/no) is part of the model and could therefore be included.

Discussion

The discussion is well conducted and explains the reasons for the inequity of access to cesarean section in Bangladesh although some assertions about Bangladesh would need to be supported by references for example: “Private health facilities in Bangladesh are mostly profit ambitious and located in urban centre”.

Conclusion

The authors should also show among the perspectives the necessity to make an assessment concerning the indications of caesarean sections.

Reviewer #3: It is a cross-sectional cohort study. The authors establish the association of the social characteristics of women about the use of cesarean section (CS) in Bangladesh; from a multi-stage survey, they were determined in the periods of 2004, 2007, 2011 , 2014 and 2017/18,

The analysis is under what is proposed in general.

In general terms, well written that part of a valid research question, and it is possible to establish how sociodemographic differences affect the use of the cesarean section, increasing inequalities.

I want the authors to clarify the hierarchical levels that were taken into account for the multilevel analysis.

6. PLOS authors have the option to publish the peer review history of their article (what does this mean?). If published, this will include your full peer review and any attached files.

**Do you want your identity to be public for this peer review?** For information about this choice, including consent withdrawal, please see our Privacy Policy.

Reviewer #1: No

Reviewer #2: No

Reviewer #3: No

---

## [Editor Report · Decision Letter 1]

7 Dec 2021

Too many yet too few caesarean section deliveries in Bangladesh: evidence from Bangladesh Demographic and Health Surveys Data

PGPH-D-21-00594R1

Dear Dr. Khan,

We're pleased to inform you that your manuscript has been judged scientifically suitable for publication and will be formally accepted for publication once it meets all outstanding technical requirements.

Within one week, you'll receive an e-mail detailing the required amendments. When these have been addressed, you'll receive a formal acceptance letter and your manuscript will be scheduled for publication.

An invoice for payment will follow shortly after the formal acceptance. To ensure an efficient process, please log into Editorial Manager at https://www.editorialmanager.com/pgph/ click the 'Update My Information' link at the top of the page, and double check that your user information is up-to-date. If you have any billing related questions, please contact our Author Billing department directly at authorbilling@plos.org.

Kind regards,

Javier H Eslava-Schmalbach, M.D., Ph.D., MSc

Academic Editor
